# The Use of SGLT-2 Inhibitors and GLP-1RA in Frail Older People with Diabetes: A Personalised Approach Is Required

**DOI:** 10.3390/metabo15010049

**Published:** 2025-01-14

**Authors:** Alan J. Sinclair, Ahmed H. Abdelhafiz

**Affiliations:** 1King’s College, London WC2R 2LS, UK; alan.sinclair@kcl.ac.uk; 2Foundation for Diabetes Research in Older People (fDROP), Diabetes Frail Ltd., Droitwich Spa WR9 0QH, UK; 3Department of Geriatric Medicine, Rotherham General Hospital, Moorgate Road, Rotherham S60 2UD, UK

**Keywords:** older people, diabetes, frailty, phenotype, SGLT-2 inhibitors, GLP-1RA, management

## Abstract

**Background:** Frailty is an increasingly recognised complication of diabetes in older people and should be taken into consideration in management plans, including the use of the new therapies of sodium glucose cotransporter-2 (SGLT-2) inhibitors and glucagon like peptide-1 receptor agonists (GLP-1RA). The frailty syndrome appears to span across a spectrum, from a sarcopenic obese phenotype at one end, characterised by obesity, insulin resistance, and prevalent cardiovascular risk factors, to an anorexic malnourished phenotype at the other end, characterised by significant weight loss, reduced insulin resistance, and less prevalent cardiovascular risk factors. Therefore, the use of the new therapies may not be suitable for every frail older individual with diabetes. **Objectives:** To review the characteristics and phenotype of frail older people with diabetes who should benefit from the use of SGLT-2 inhibitors or GLP-1RA. **Methods:** A narrative review of the studies investigating the benefits of SGLT-2 inhibitors and GLP-1RA in frail older people with diabetes. **Results:** The current evidence is indirect, and the literature suggests that the new therapies are effective in frail older people with diabetes and the benefit appears to be proportional with the severity of frailty. However, frail patients described in the literature who benefited from such therapy appeared to be either overweight or obese, and to have a higher prevalence of unfavourable metabolism and cardiovascular risk factors such as dyslipidaemia, gout, and hypertension compared to non-frail subjects. They also have a higher prevalence of established cardiovascular disease compared with non-frail individuals. In absolute terms, their higher cardiovascular baseline risk meant that they benefited the most from such therapy. The characteristics of this group of frail patients fulfil the criteria of the sarcopenic obese frailty phenotype, which is likely to benefit most from the new therapies due to the unfavourable metabolic profile of this phenotype. There is no current evidence to suggest the benefit of the new therapies in the anorexic malnourished phenotype, which is underrepresented or totally excluded from these studies, such as in patients living in care homes. This phenotype is likely to be intolerant to such therapy due to its associated risk of inducing further weight loss, dehydration, and hypotension. **Conclusions:** Clinicians should consider the early use of the new therapies in frail older people with diabetes who are either of normal weight, overweight, or obese with prevalent cardiovascular risk factors, and avoid their use in those frail subjects who ae underweight, anorexic, and malnourished.

## 1. Introduction

Diabetes prevalence is increasing in older people and is highest (24%) in those ≥75 years of age due to increased life expectancy [1]. Frailty is emerging as a high impact complication of diabetes in this age group [2]. Frailty prevalence is linked to old age, reaching up to 25% of people aged ≥85 years and is prevalent in around 32–48% of older people with diabetes [3]. Clinical trials have shown that the new therapies of sodium glucose cotransporter-2 (SGLT-2) inhibitors and glucagon like peptide-1 receptor agonists (GLP-1RA) to improve cardiorenal outcomes and their actions extend to include older people with diabetes [4]. For example, in a systematic review and meta-analysis, GLP-1RA and SGLT-2 inhibitors reduced major adverse cardiovascular events (MACE) in patients with diabetes ≥65 years of age. SGLT-2 inhibitors also reduced heart failure hospitalisation and composite renal endpoints. Similar results were also observed for patients with diabetes ≥75 years old [5]. There is no direct evidence of the benefits of these new therapies in frail older people, as this population was largely excluded from the clinical trials and data are extrapolated from their efficacy in biologically well older people with diabetes. Therefore, there is a clinical inertia for the use of such therapies in this category of patients [6]. The current literature encourages the use of the new therapies in frail older people with diabetes but without providing clear recommendations on the characteristics of patients who are likely to benefit from such therapies [7,8,9]. Frailty induces body composition and metabolic changes, which may have an impact on the choice of the antidiabetic therapy according to frailty metabolic phenotype [10]. Therefore, this manuscript reviews the metabolic changes associated with frailty and describes the current evidence to characterise frail older people with diabetes who should benefit from the new therapies. This may help clinicians to personalise the use of the new therapies in this vulnerable group of patients and to address the current gap in the literature.

## 2. Methods

The literature search involved three databases–Google Scholar, Medline, and Embase–covering articles published from 1996 to present. We employed Medical Subject Heading (MeSH) terms such as “older people”, “elderly”, “aged”, “frail”, “frailty”, “frailty status”, “diabetes mellitus”, “SGLT-2 inhibitors”, “GLP-1RA”, “new therapies”, “hypoglycaemic therapy”, “anti-diabetic therapy”, “cardiovascular risk”, “evidence base”, “interventions”, and “clinical trials”. Our initial search utilised the following MeSH terms: “older” OR “elderly” OR “aged” AND “frail” OR “frailty” AND “type 2 diabetes mellitus”. Subsequently, we expanded our search by incorporating therapy-related terms (e.g., SGLT-2 inhibitors, GLP-1RA, new therapies, hypoglycaemic therapy, anti-diabetic therapy) using similar search strategies. We supplemented our search with additional keywords such as “cardiovascular risk”, “interventions”, and “outcomes”, utilising individual and combined terms. We limited our selection to articles published in the English language. Articles were initially screened for relevance based on their abstracts. Furthermore, we conducted a manual review of citations within retrieved articles to identify additional relevant studies beyond those captured in the electronic searches. We included clinical studies that reported outcomes of SGLT-2 inhibitors or GLP-1RA in frail older people with diabetes as well as the characteristics of these patients. The authors independently reviewed all the studies identified through the search process. For each study, data were extracted in tables including a description of patients’ characteristics and outcomes in frail compared with non-frail individuals.

## 3. Frailty and Diabetes

In addition to the traditional micro and macrovascular diseases, frailty is emerging as a new third category of diabetes-related complication [2]. Frailty is likely to be caused by the progressive accumulation of subclinical damage in multiple organ systems, therefore diabetes-related complications may lead to frailty. It has been shown that diabetes-related complications and diabetes-associated comorbidities increase the risk of frailty and, in turn, frailty increases the risk of diabetes in a reciprocal manner [11]. In a Japanese study, the risk of frailty increased in individuals with a history of diabetes and impaired renal function {odds ratio (OR) 2.76, 95% confidence interval (CI) 1.21 to 8.24} [12]. The Mexican health and nutrition survey of 7164 older people, mean (SD) age 70.6 (8.1) years, reported the association of diabetes with frailty (coefficient 0.28, *p* < 0.001) and the association was incremental when hypertension (0.63, *p* < 0.001) or any diabetes-related complication was also present (0.55, *p* < 0.001) [13]. It is likely that diabetes-related vascular complications are associated with inactivity and physical and cognitive decline, suggesting that frailty may be the result of diabetes-induced end-organ damage [14,15].

Both hyper and hypoglycaemia increase the risk of frailty [16,17]. Therefore, glycaemia and frailty appear to have a U-shaped relationship [18]. Persistent hyperglycaemia causes frailty by increasing the prevalence of vascular complications and inducing low-grade inflammation, oxidative stress, and mitochondrial dysfunction [19,20,21]. Hypoglycaemia causes frailty by the recurrent variations in blood glucose levels, which may induce endothelial damage [22]. Hypoglycaemia is also prevalent in conditions such as dementia and cerebrovascular disease, which are associated with frailty [23,24]. In a reciprocal manner, frailty and pre-frailty increased the risk of incident type 2 diabetes mellitus, which may be due to insulin resistance and glucose dysregulation [25,26].

## 4. Frailty Metabolic Spectrum

Because of body composition changes, frailty is likely to span across a wide metabolic spectrum, which will affect diabetes management. For example, frailty is associated with sarcopenia or loss of skeletal muscle mass [27]. Skeletal muscle is the major site of glucose uptake and a reduction in muscle mass will increase insulin resistance. On the other hand, significant weight loss will reduce insulin resistance and the need for antidiabetic medications. Although weight loss is one of the criteria of frailty, it is not a mandatory requisite for frailty diagnosis and obese individuals can be frail [27]. Obesity, especially visceral fat deposition, accompanied with sarcopenia will constitute one end of the metabolic spectrum, which is characterised by increased insulin resistance. This sarcopenic obese (SO) phenotype will have an overall unfavourable metabolic profile similar to the metabolic syndrome such as dyslipidaemia, hypertension, and progressive hyperglycaemic trajectory, which requires an intensification of therapy [28]. The metabolic spectrum of frailty gradually spans from the SO phenotype, across frail individuals with varying degrees of body weights, reaching the anorexic malnourished phenotype (AM) at the other end of the spectrum. The AM phenotype is characterised by significant weight loss, reduced insulin resistance, and regressive hyperglycaemic trajectory. The cardiovascular (CV) risk factors decline in a state of reverse metabolism, which will require a deintensification of therapy. Therefore, frailty includes both low and high body mass index individuals, in a U-shaped relationship across a span of different metabolic characteristics. The metabolic characteristics depend on body weight, the varying muscle mass/visceral fat ratios, and the corresponding varying degrees of insulin resistance [29]. This U-shaped relationship between BMI and frailty is detrimental in mortality as an outcome. The mortality in the SO phenotype with a higher BMI and prevalent CV risk factors is likely to be related to CV disease. The mortality in the AM with lower BMI and less prevalent CV risk factors is likely to be related to malnutrition (Figure 1).

## 5. The New Therapies in Frail Older People

The clinical trials of the new therapies showed efficacy and safety across all age groups, including those ≥65 years of age [4]. However, there is a lack of data on very old subjects >75 years old and, in addition, frailty was not assessed in these trials. Therefore, there is little direct evidence derived from clinical trials on the use of the new therapies in frail older people with diabetes. However, there is recently published indirect evidence from observational studies. The largest is the retrospective cohort study by Kutz et al., which included a total of 744,310 older patients ≥65 years of age with type 2 diabetes mellitus, to compare the cardiovascular effectiveness and safety of SGLT-2 inhibitors and GLP-1RA according to frailty status. The study found that the largest absolute benefit was among the frail group over a mean (SD) follow-up of 10.6 (11.3) months [30]. The study included three 1:1 propensity score–matched cohorts, each stratified by three frailty strata (non-frail, pre-frail, and frail), using data from the Medicare beneficiaries (2013–2019) including patients with type 2 diabetes who initiated SGLT-2 inhibitors, GLP-1RA, or dipeptidyl peptidase-4 (DPP-4) inhibitors. The primary outcome of CV benefit was a composite of acute myocardial infarction, ischaemic stroke, hospitalisation for heart failure, and all-cause mortality. The primary outcome of safety was a composite of severe drug-related adverse events. The primary CV outcome benefit of SGLT-2 inhibitors was 0.72 (95% CI 0.69 to 0.75). There was a larger absolute rate reduction among frail people, with an incidence rate difference (IRD) of −27.24 (−41.64 to −12.84) in frail and −6.74 (−8.61 to −4.87) in non-frail people compared with DPP-4 inhibitors. Similarly, the primary CV outcome benefit of GLP-1RA was 0.74 (0.71 to 0.77), with an IRD of −25.88 (−38.30 to −13.46) in frail and −7.02 (−9.23 to −4.81) in non-frail people compared with DPP-4 inhibitors. Although the relative risk reduction for the effectiveness outcomes remained stable across frailty strata, the absolute reductions were higher in the frailer population, reflecting their higher baseline risk for these outcomes. Therefore, frailer older people with diabetes will experience larger benefits from the new therapies than those without frailty. The number needed to treat (NNT) to prevent a CV event over one year was 39 vs. 159 for SGLT-2 inhibitors and 42 vs. 162 for GLP-1RA compared with DPP-4 inhibitors. The primary outcome of safety was comparable between SGLT-2 inhibitors, GLP-1RA, and DPP-4 inhibitors. However, frail people initiating SGLT-2 inhibitors had a larger absolute rate increase in genital infection but a lower risk for heart failure hospitalisation compared to both GLP-1RA and DPP-4 inhibitors [30]. The Dapagliflozin Evaluation to Improve the Lives of Patients with Preserved Ejection Fraction Heart Failure (DELIVER) study examined the effects of the SGLT-2 inhibitor dapagliflozin according to frailty status. Although the study was not designed exclusively for people with diabetes, most of the frail subjects (72.5%) had diabetes mellitus. The study showed the benefits of dapagliflozin to be greater in patients with higher levels of frailty. The effects of dapagliflozin (hazard ratio) on the primary end point (time to a first worsening heart failure event or CV death) were 0.85 (95% CI 0.68 to 1.06), 0.89 (0.74 to 1.08), and 0.74 (0.61 to 0.91) in frail, more frail, and most frail subjects, respectively. Adverse reactions and treatment discontinuation were not more common with dapagliflozin than with a placebo, irrespective of the frailty class [31]. The study included a total of 6263 subjects >40 years of age and the frailty index was calculable in 6258 subjects. The overall relative risk reduction was 18% to each frailty class compared with placebo. This results in a NNT of 40 non-frail, 31 more frail, and 19 most frail, respectively, to prevent one primary event over the median follow-up of 2.3 years. The improvement in symptom burden and quality of life with dapagliflozin appeared as early as four months of treatment and was more significant in patients with greater frailty [31]. In the post-hoc analysis of the Dapagliflozin and Prevention of Adverse Outcomes in Heart Failure (DAPA-HF) trial, dapagliflozin reduced the risk of CV mortality or worsening HF regardless of frailty status and the effects were better the worse the frailty state was. The differences in event rate per 100-person-years for dapagliflozin versus placebo from lowest to highest frailty class were −3.5 (95% CI −5.7 to −1.2) in non-frail, −3.6 (−6.6 to −0.5) in more frail, and −7.9 (−13.9 to −1.9) in most frail subjects after a median follow-up of 18.2 months. Consistent benefits were observed for other clinical events and health status and the absolute reductions were larger in the most frail patients. The NNT to prevent one event per 100-person-years was 31, 25, and 15, respectively, in the lowest to the highest frailty class [32].

## 6. Patients’ Characteristics

Patients’ characteristics of the above studies are summarised in Table 1, Table 2 and Table 3. Although weight loss is one of the criteria of frailty [27], most of the frail cohorts included in the study by Kutz et al. were overweight or obese (61.68% vs. 39.01% for SGLT-2 inhibitors cohort) and 66.54% vs. 44.60% for the GLP-1RA cohort) compared to non-frail subjects. In addition, obesity-related clinical conditions were more common in frail compared to non-frail groups, such as obstructive sleep apnea (31.75% vs. 11.07% for SGLT-2 inhibitors and 35.76% vs. 14.03 for GLP-1RA) and gastroesophageal reflux disease (48.56% vs. 17.80% and 49.20% vs. 17.60%, respectively) [30]. This suggests that the frail group were not incidentally obese but have established obesity-related complications [33]. Moreover, the frailty cohort in both the new therapy cohorts showed an unfavourable metabolic profile, such as increased prevalence of hyperlipidaemia, hypertension, chronic kidney disease (CKD), and fatty liver. In addition, they had a more prevalent history of CV events compared to non-frail cohorts. As a result, their mean combined comorbidity score was higher compared with the non-frail cohort. Although age was comparable in the frail and non-frail individuals, there was a female predominance in the frail cohorts, which reflects the higher risk of CV disease in women compared to men with diabetes [34]. (Table 1a,b) Furthermore, the study appears to include mildly frail subjects. The claims-based frailty index (CFI) was used to measure frailty, which estimates a deficit accumulation frailty index (range 0–1) [35]. In this study, the frailty status was defined as non-frail, CFI < 0.15, pre-frail, CFI 0.15–0.24, and frail, CFI ≥ 0.25. The CFI categories according to severity of frailty are robust <0.15, pre-frail 0.15–0.24, mildly frail 0.25–0.34, and moderate-to-severely frail ≥0.35) [36]. The mean (SD) of CFI of the frail cohorts in both SGLT-2 inhibitors and GLP-1RA was only 0.28 (0.03), which is in the lower end of the mild frailty category, suggesting a low grade of frailty [30]. In addition, this study excluded care home residents or even those who had had an episode of skilled nursing home admission in the previous 365 days to the entry date. Care home residents are likely to be predominantly anorexic with significant weight loss, as the process of institutionalisation itself increases the risk of malnutrition and the prevalence of malnutrition is frequently high in these settings [37]. Similarly, in the DELIVER study, frail people were significantly more obese (BMI 32.1 vs. 28.1) and had significantly more prevalent obesity-related clinical conditions, such as obstructive sleep apnea (17.5% vs. 2.4%), compared with non-frail individuals. In addition, frail subjects were older, have significantly more prevalent CV risk factors, CV complications, and CV events than non-frail counterparts [31]. (Table 2) The DELIVER study is limited by the exclusion of patients with the greatest level of frailty, and it is likely that the participants in the study were less frail than patients in the general population. Other frailty scores that include assessments of muscle strength and functional capacity were not measured. In the DAPA-HF study, most frail subjects were older and more likely to have CV and non-CV comorbidities compared with non-frail subjects. They had a higher body mass index, obstructive sleep apnoea, CKD, dyslipidaemia, peripheral vascular disease, and worse heart failure. This study was limited by the exclusion of very high-risk patients and the lack of direct measurement of muscle strength or functional capacity [32].

## 7. Clinical Implications

It appears from the little available evidence that the new therapies are beneficial in frail patients with at least normal body weight. Frail overweight or obese patients with the SO phenotype, at the end of the spectrum, stand to gain the most benefit. The skeletal muscles secret anti-inflammatory myokines, which decrease with sarcopenia, while adipose tissue secretes pro-inflammatory adipokines, which increases with obesity. Therefore, the SO frailty phenotype is associated with a reduced myokines/adipokines ratio. This leads to an unfavourable metabolism, which further increases the risk of diabetes-associated CV disease by promoting chronic low-grade inflammation, increasing oxidative stress and mitochondrial dysfunction [38]. The SO phenotype will have a higher baseline CV risk, and will therefore benefit most from the new therapies with the highest risk reduction and lowest NNT compared to non-frail subjects. Although the current evidence is not substantial, the study by Kutz et al. was a large study, which included a total of 38,272 frail older people with diabetes. Other evidence can be drawn from the secondary analysis of the VERTIS CV study, which examined the effects of ertugliflozin on cardiorenal outcomes, kidney function, and safety outcomes, and showed that the effects were generally similar across age subgroups, including those ≥75 years old. The study did not assess for frailty, but the mean (SD) weight of people aged ≥75 years was 84.7 (16.9) Kg and BMI 30.2 (5.0), indicating overweight/obesity. In addition, the mean (SD) HbA1c was 8.0% (0.9) and subjects have more baseline cardiorenal complications than patients <65 years old [39]. The post-hoc analysis of the DECLARE–TIMI 58 study, which included 1096 (6.4%) subjects ≥75 years old, concluded an overall safety and efficacy of dapagliflozin across all age groups, although it did not assess for frailty. However, this older age group had a median (IQR) BMI 30.2 (27.4, 33.9), HbA1c 7.8 (7.2, 8.5), longer duration of diabetes, and higher prevalence of cardiorenal complications than younger people [40]. A retrospective study, which included 235 very elderly patients with type 2 diabetes, with a mean (SD) age of 79.6 (3.9) years, found SGLT-2 inhibitors to be safe and well-tolerated. The study did not assess for frailty but 44.3% of the participants were obese and the mean (SD) HbA1c was 7.9% (1.4) [41]. The prospective observational SGLT-2 inhibitors in older diabetic patients (SOLD) study, which included 739 older people with a mean (SD) age of 75.4 (3.9) years, found SGLT-2 inhibitors to be a safe and effective therapeutic option in this age group of older patients. However, 82.7% of the subjects were either obese or overweight and the mean (SD) HbA1c was 7.9% (1.1) [42]. It has recently been shown that obese frail older people with diabetes, median {interquartile range (IQR)} BMI 33.5 (29.5, 38.6), have a higher CV risk than non-frail, median (IRQ) BMI 29.5 (26.8, 33.2), participants, with a hazard ratio (95% CI) of 1.70 (1.53 to 1.90) [43]. This suggests that the overweight/obese side of the frailty spectrum is likely to be at higher risk of CV events than the underweight side of the spectrum [44]. Because of the above results, the new therapies, combined with an intensification of CV risk factor treatment, should be initiated early in frail older people with normal or higher body weight and especially in the SO frail phenotype, which will benefit most from such therapies (Figure 2). However, in the SO phenotype, the use of SGT-2 inhibitors may lead to further sarcopenia despite the favourable effect on body fat and body weight [45]. Therefore, such therapies should be combined with resistance exercise training and adequate diet to preserve muscle mass and reduce the progression of frailty. In addition, due to the risk of SGLT-2 inhibitor-related side effects, this therapy should be temporarily withheld in acute medical conditions leading to hospitalisation or the need of surgical interventions.

The new therapies will not be suitable for frail older people with lower body weight, especially those at the end of the frailty spectrum of the AM phenotype. This phenotype is characterised by anorexia and significant weight loss and the new therapies, by inducing further anorexia and weight loss, will not be suitable. In the above-mentioned retrospective study, fourteen adverse reactions were attributed to cachexia, urinary frequency, and weight loss, although the demographic characteristics of these fourteen cases were not reported [41]. In addition, this phenotype, due to reduced oral intake, will also be at risk of dehydration. The safety outcome in the study by Kutz et al. was a composite of lower-limb amputation, nonvertebral fracture, hospital or emergency department admission for hypoglycaemia, hospitalisation for acute kidney injury, diabetic ketoacidosis, severe genital or urinary tract infection, acute pancreatitis, or a non-malignant biliary event. It did not, however, include important adverse events relevant to frail older people such as dehydration, hypotension, anorexia, weight loss, and falls [30]. After one year of follow-up in the SOLD study, treatment was withdrawn in 174 (23.5%) patients. Genitourinary infections were the most common cause of treatment withdrawal, 44.1% at 6 months and 41.7% at 12 months, followed by intolerance (excessive diuresis, nausea, lack of appetite) in 16.6% at 6 months and 20.8% at 12 months. Acute kidney injury occurred in 0.8% at 6 months and 12.5% at 12 months. Volume depletion (hypotension, orthostatic hypotension, pre-syncope, and syncope) in 11% occurred only in the first 6 months. It was noted that the patients who discontinued treatment tended to have lower BMI values of 27.9 (3.3) vs. 29.2 (4.7), *p* = 0.001, and were of an older age, mean (SD) 75.8 (4.2) vs. 74.7 (3.8) years, *p* = 0.002. Discontinuation was almost two times higher in patients aged ≥80 years, compared to younger patients (35% vs. 19.1%, *p* = 0.005). In addition, lower BMI values at baseline were significantly related to SGLT-2 inhibitors suspension, OR 0.92, 95% CI 0.88 to 0.97, *p* < 0.001 [42]. This supports the evidence that the new therapies are likely to be less tolerated in frail patients with low body weight/BMI, especially if they are anorexic at the end of the frailty metabolic spectrum (AM phenotype). The AM phenotype is likely to include older people living in care homes, who were excluded in the study by Kutz et al. [30]. The prevalence of anorexia is associated with ageing and frailty, and the prevalence is highest (34.1%) in care homes settings [46]. It has also been shown that BMI, free fat mass, and fat mass are significantly lower in residents of care homes who have malnutrition, sarcopenia, and frailty [47]. Because of significant weight loss in the AM phenotype, the insulin requirement decreases, the diabetes trajectory decelerates, and CV risk factors decline, including insulin resistance, in a state of a reverse metabolism [48]. Weight loss improves insulin sensitivity in internal organs such as the liver and the skeletal muscles and reduces fat deposition in the pancreas, which enhances β-cells insulin secretion [49,50]. This may lead to spontaneous resolution of hyperglycaemia and normalisation of HbA1c in some AM phenotype patients [51]. Therefore, early initiation of long-acting insulin analogues may be suitable, if therapy is required, in this frailty phenotype due to its anabolic and weight-gaining properties. The lower risk of hypoglycaemia of the analogues compared with NPH insulins makes it more suitable in frail older people [52]. The advances in continuous glucose monitoring are crucial in glucose control and reducing the risk of hypoglycaemia. The newly developed microneedle sensors, as minimally invasive devices, offer a real-time sensing patch, which is painless, flexible, and can be used at point-of-care settings [53]. However, regular assessments and deintensification of therapy should be considered in this AM frailty phenotype as they continue to lose weight. In addition, individualised nutritional and hydration strategies in a holistic approach should be emphasised to avoid further weight loss and progression of frailty. The focus of care should be on preventing extreme dysglycaemia and unnecessarily hospitalisation, and on the maintenance of quality of life (Figure 2).

## 8. Conclusions

This review fills an important gap in our understanding of metabolic phenotypes of frailty and guides clinicians in their daily decision-making for the use of the new therapies in the high-risk group of patients with frailty. Although frailty is an important emerging complication of diabetes in older people, there is no direct evidence to suggest the benefits of the new therapies in this group of patients. The current available evidence from recent studies is indirect and suggests that the new therapies are beneficial in frail older people with diabetes [30,31,32]. However, the conclusion in the literature that the new therapies are useful in frail older people with diabetes, seen as one homogenous group of patients, is inaccurate. Frail people in these studies were mostly either overweight or obese, suggesting that the SO end of the metabolic spectrum of frailty are the ones who should benefit from such therapies. The unfavourable metabolic profile of this phenotype and the highly prevalent CV risk factors place this group of patients at a high baseline CV risk and therefore mean that they are most likely to benefit from such therapies. There is no literature to suggest benefits of the new therapies in the AM frailty phenotype. The recent studies either excluded or hardly included patients living in care homes, who are likely to be AM frail. This frailty phenotype is likely to be intolerant to such therapies due to the risk of inducing further weight loss, dehydration, and hypotension. This review clarifies this confusion in the literature and highlights the importance of taking into consideration the metabolic heterogeneity of frailty. Therefore, clinicians should consider the early use of the new therapies in frail older people with diabetes who are normal or overweight with prevalent CV risk factors and be careful in the underweight or AM frail phenotype. Since the proportion of older people will continue to increase, there is a need for clinical trials to be more representative of the heterogeneity of this age group. This includes further exploring frailty metabolic diversity and investigating therapeutic outcomes relevant to older people with diabetes, such as functional status, cognitive performance, and quality of life.

## 9. Future Perspectives

The metabolic impact of frailty on diabetes has not yet been fully explored. Frailty is likely to span across a metabolic spectrum with variable metabolic effects. These range from a sarcopenic obese frail phenotype, which benefits from the CV risk reduction properties of the new therapies and an anorexic malnourished frail phenotype, which is likely not suitable for such therapy. Therefore, future long-term CV clinical trials should assess the metabolic characteristics of frail older people with type 2 diabetes from the outset to prospectively confirm the risk-benefit across different frailty phenotypes. The cause of frailty is likely to be related to diabetes-related renal and cardiovascular complications, which are reduced by the new therapies. Therefore, these protective effects of the new therapies may lead to a reduction in incident frailty and will need to be further explored. Although the new therapies are associated with weight loss and a reduction in visceral fat, they may also reduce muscle mass and worsen sarcopenia. Therefore, a combination of the new therapies with insulin, as a potential muscle-building agent, will need future investigation. Similarly, the weight-gaining and anabolic properties of insulin need to be explored in the AM frailty phenotype. There is also a specific need for future research to address the gaps in evidence for the AM frailty phenotype and to develop tailored therapeutic approaches that prioritise both metabolic and functional outcomes in this population. The new therapies have several extra glycaemic effects, and, therefore, novel therapies with extra glycaemic effects on frailty, such as improving muscle function and muscle strength, are required. In addition, the direct effect of glycaemic control on the incidence of frailty requires further exploration.

## 10. Key Points

Frailty in older people with diabetes is a heterogeneous condition with a wide metabolic spectrum.The current literature suggests that the new therapies are beneficial in frail older people with diabetes but it inaccurately refers to frailty as one homogenous group.The evidence was shown only in frail older people, who were overweight or obese, suggesting that the benefit is likely to be most significant in the sarcopenic obese (SO) frail phenotype, which has an unfavourable metabolic profile.There is no evidence to suggest the benefits of the new therapies in the anorexic malnourished (AM) frailty phenotype or care home residents, who were largely excluded from the studies.Future studies are still required to further characterise frail older people with diabetes from the outset to prospectively confirm the risk-benefit across different frailty phenotypes.

## Figures and Tables

**Figure 1 metabolites-15-00049-f001:**
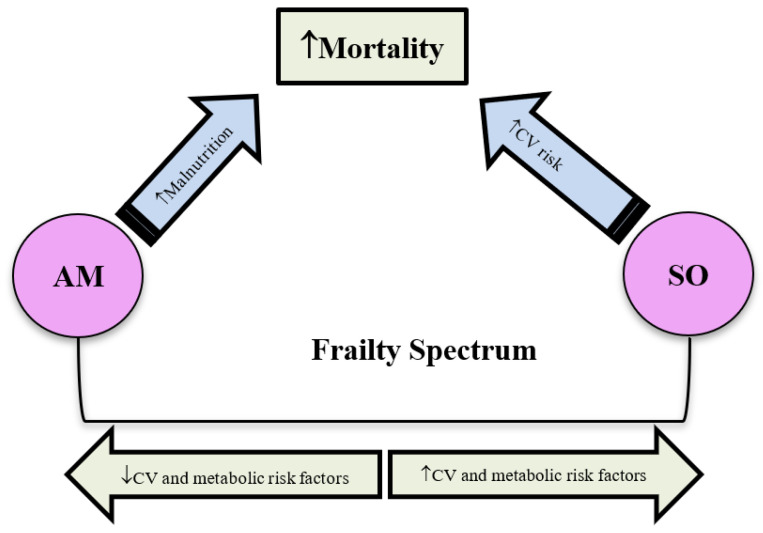
Metabolic spectrum of frailty with a SO phenotype at one end, characterised by exaggerated CV risk factors, and AM phenotype at the other end, characterised by regression of CV risk factors. CV = Cardiovascular, AM = Anorexic malnourished, SO = Sarcopenic obese.

**Figure 2 metabolites-15-00049-f002:**
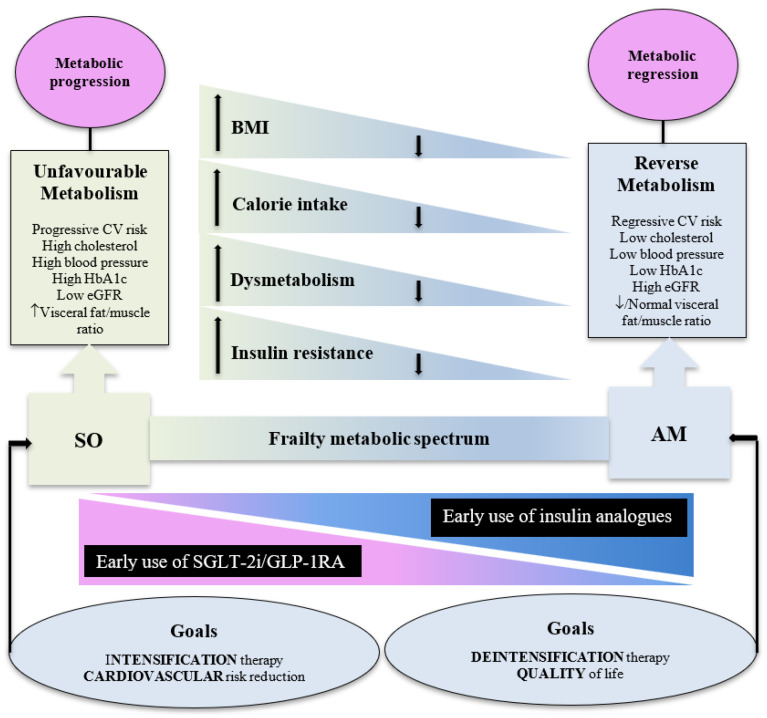
Suggested criteria of frail older people with diabetes who should benefit from the early initiation of the new therapies of SGLT-2 inhibitors and GLP-1RA. Frailty is likely to span across a metabolic spectrum that ranges from an SO phenotype on the one side to an AM on the other. Early introduction of the new therapies is indicated in the SO phenotype and should be less used gradually as the BMI declines, reaching the other end of the spectrum at the AM phenotype when early introduction of insulin analogues is appropriate. BMI = Body mass index, eGFR = Estimated glomerular filtration rate, SO = Sarcopenic obese, AM = Anorexic malnourished, SGLT-2i = Sodium glucose cotransporter-2 inhibitors, GLP-1RA = glucagon-like peptide-1 receptor agonists.

**Table 1 metabolites-15-00049-t001:** (**a**): Metabolic profile of frail vs. non-frail older people with diabetes initiated on SGLT-2 inhibitors and propensity score–matched to patients treated with DPP-4 inhibitors [30]. (**b**): Metabolic profile of frail vs. non-frail older people with diabetes initiated on GLP-1RA and propensity score–matched to patients treated with DPP-4 inhibitors [30].

(a)		
Characteristics	Frail (N = 5710)	Non-Frail (N = 50,843)
Age, mean (SD)	72.83 (5.63)	71.33 (4.66)
Gender, female (%)	3627 (63.52)	20,246 (39.82)
Mean (SD) CFI	0.28 (0.03)	0.13 (0.02)
Mean (SD) CCS	5.59 (2.88)	0.74 (1.54)
Overweight (%)	693 (12.14)	4837 (9.51)
Obesity (%)	2829 (49.54)	15,000 (29.50)
OSA (%)	1813 (31.75)	5628 (11.07)
GORD	2773 (48.56)	9050 (17.80)
Hyperlipidaemia (%)	5128 (89.81)	42,054 (82.71)
Hypertension (%)	5631 (98.62)	43,826 (86.20)
Coronary atherosclerosis (%)	3759 (65.83)	7401 (14.56)
Unstable angina (%)	489 (8.56)	326 (0.64)
Acute MI (%)	385 (6.74)	248 (0.49)
AF (%)	1705 (29.86)	2265 (4.45)
Heart failure (%)	2309 (40.44)	1257 (2.47)
Cardiomyopathy (%)	538 (9.42)	672 (1.32)
Other cardiovascular disease (%)	2904 (50.86)	3664 (7.21)
Coronary procedure (%)	278 (4.87)	247 (0.49)
History of CABG or PCI (%)	1689 (29.58)	2242 (4.41)
Cerebral atherosclerosis (%)	593 (10.39)	214 (0.42)
Transient ischemic attack (%)	582 (10.19)	397 (0.78)
Ischemic stroke (%)	1882 (32.96)	2046 (4.02)
Other cerebrovascular disease (%)	1575 (27.58)	887 (1.74)
Cerebrovascular procedure (%)	41 (0.72)	21 (0.04)
PVD (%)	1737 (30.42)	3111 (6.12)
Lower limb amputation (%)	98 (1.72)	111 (0.22)
Other atherosclerosis (%)	179 (3.13)	266 (0.52)
CKD (%)	1928 (33.77)	3982 (7.83)
Hypertensive nephropathy (%)	1225 (21.45)	1508 (2.97)
NASH or NAFLD (%)	484 (8.48)	2321 (4.57)
**(b)**		
**Characteristics**	**Frail (N = 8474)**	**Non-Frail (N = 39,675)**
Age, mean (SD)	73.92 (6.35)	70.93 (4.47)
Gender, female (%)	5795 (68.39)	17,764 (44.77)
Mean (SD) CFI	0.28 (0.03)	0.13 (0.02)
Mean (SD) CCS	5.87 (2.91)	0.89 (1.61)
Overweight (%)	809 (9.55)	3152 (7.94)
Obesity (%)	4829 (56.99)	14,724 (37.11)
OSA (%)	3030 (35.76)	5567 (14.03)
GORD	4169 (49.20)	6984 (17.60)
Hyperlipidaemia (%)	7583 (89.49)	32,866 (82.84)
Hypertension (%)	8364 (98.70)	34,539 (87.05)
Coronary atherosclerosis (%)	5315 (62.72)	4928 (12.42)
Unstable angina (%)	643 (7.59)	220 (0.55)
Acute MI (%)	528 (6.23)	137 (0.35)
AF (%)	2297 (27.11)	1562 (3.94)
Heart failure (%)	3616 (42.67)	947 (2.39)
Cardiomyopathy (%)	731 (8.63)	438 (1.10)
Other cardiovascular disease (%)	4263 (50.31)	2802 (7.06)
Coronary procedure (%)	303 (3.58)	149 (0.38)
History of CABG or PCI (%)	2177 (25.69)	1427 (3.60)
Cerebral atherosclerosis (%)	846 (9.98)	158 (0.40)
Transient ischemic attack (%)	752 (8.87)	301 (0.76)
Ischemic stroke (%)	2666 (31.46)	1494 (3.77)
Other cerebrovascular disease (%)	2304 (27.19)	692 (1.74)
Cerebrovascular procedure (%)	53 (0.63)	10 (0.03)
PVD (%)	2717 (32.06)	2541 (6.40)
Lower limb amputation (%)	191 (2.25)	132 (0.33)
Other atherosclerosis (%)	261 (3.08)	224 (0.56)
CKD (%)	3785 (44.67)	5309 (13.38)
Hypertensive nephropathy (%)	2462 (29.05)	2131 (5.37)
NASH or NAFLD (%)	671 (7.92)	1927 (4.86)

SGLT-2 = Sodium glucose cotransporter, DPP-4 = Dipeptidyl peptidase, N = Number, SD = Standard deviation, CFI = Claims-based frailty index, CCS = Combined comorbidity score, OSA = Obstructive sleep apnea, GORD = Gastroesophageal reflux disease, MI = Myocardial infarction, AF = Atrial fibrillation, CABG = Coronary artery by-pass grafting, PCI = Per-cutaneous coronary intervention, PVD = Peripheral vascular disease, CKD = Chronic kidney disease, NASH = Non-alcoholic steatohepatitis, NAFLD = Non-alcoholic fatty liver disease. GLP-1RA = Glucagon-like peptide receptor agonists.

**Table 2 metabolites-15-00049-t002:** Metabolic profile of most frail vs. non-frail older people with heart failure treated with the SGLT-2 inhibitor dapagliflozin [31].

Characteristics	Most Frail (N = 1491)	Non-Frail (N = 2354)	*p* Value
Mean (SD) age	72.7 (8.8)	70.1 (10.3)	<0.001
Age ≥ 76 years (%)	593 (39.8)	785 (33.3)	<0.001
Gender, male (%)	841 (56.4)	1308 (55.6)	0.79
FI	≥0.311	≤0.210	
Mean (SD) HbA1c	7.1% (1.6)	6.2% (1.2)	<0.001
Mean (SD) BMI	32.1 (6.2)	28.1 (5.8)	<0.001
Mean (SD) creatinine, μmol/L	117.3 (34.8)	91.1 (24.2)	<0.001
Mean (SD) eGFR, mL/min/1.73 m^2^	52.1 (17.4)	68.7 (18.0)	<0.001
eGFR < 60 mL/min/1.73 m^2^ (%)	1070 (71.8)	697 (29.6)	<0.001
Type 2 DM (%)	1081 (72.5)	558 (23.7)	<0.001
OSA (%)	261 (17.5)	57 (2.4)	<0.001
Dyslipidaemia (%)	1294 (86.8)	969 (41.2)	<0.001
Hypertension (%)	1459 (97.9)	1814 (77.1)	<0.001
AF (%)	976 (65.5)	1188 (50.5)	<0.001
Angina (%)	678 (45.5)	227 (9.6)	<0.001
MI (%)	643 (43.1)	319 (13.6)	<0.001
CABG/PCI (%)	855 (57.3)	381 (16.2)	<0.001
Heart failure hospitalisation (%)	750 (50.3)	821 (34.9)	<0.001
Heart failure > 5 years (%)	526 (35.3)	534 (22.7)	<0.001
NYHA class III/IV (%)	535 (35.8)	410 (17.4)	<0.001
Any coronary artery disease	1146 (76.9)	694 (29.5)	<0.001
Stroke (%)	280 (18.8)	92 (3.9)	<0.001
Stroke/TIA (%)	363 (24.3)	115 (4.9)	<0.001
PVD (%)	278 (18.6)	44 (1.9)	<0.001
Any atherosclerosis (%)	1250 (83.8)	812 (34.5)	<0.001
Non-coronary revascularisation (%)	81 (5.4)	7 (0.3)	<0.001
Gout (%)	287 (19.2)	89 (3.8)	<0.001

SGLT-2 = Sodium glucose transporter, N = Number, SD = Standard deviation, FI = Frailty index, BMI = Body mass index, eGFR = Estimated glomerular filtration rate, DM = Diabetes mellitus, OSA = Obstructive sleep apnea, AF = Atrial fibrillation, MI = Myocardial infarction, CABG = Coronary artery by-pass grafting, PCI = Per-cutaneous coronary intervention, NYHA = New York heart association, TIA = Transient iscahaemic attack, PVD = Peripheral vascular disease.

**Table 3 metabolites-15-00049-t003:** Metabolic profile of non-frail, more frail, and most frail older people with heart failure treated with the SGLT-2 inhibitor dapagliflozin (post-hoc analysis) [32].

Characteristics	Non-Frail (N = 2392)	More Frail (N = 1606)	Most Frail (N = 744)
Mean (SD) age	63.6 (11.6)	68.8 (9.4)	69.8 (9.0)
Gender, male (%)	1844 (77.1)	1225 (76.3)	564 (75.8)
FI	≤0.210	0.211–0.310	≥0.311
Median (IQR) HbA1c	5.9 (5.6–6.4)	6.2 (5.8–7.0)	6.7 (6.0–7.7)
Mean (SD) BMI	26.9 (5.7)	28.9 (5.8)	30.6 (6.1)
Mean (SD) creatinine, μmol/L	94.7 (34.8)	109.8 (30.8)	124.2 (36.4)
Mean (SD) eGFR, mL/min/1.73 m^2^	73.0 (18.4)	60.9 (17.4)	53.0 (16.6)
eGFR < 60 mL/min/1.73 m^2^ (%)	568 (23.7)	831 (51.7)	527 (70.8)
Type 2 DM (%)	694 (29.0)	882 (54.9)	563 (75.7)
OSA (%)	57 (2.4)	88 (5.5)	125 (16.8)
Dyslipidaemia (%)	1024 (42.8)	1185 (73.8)	660 (88.7)
Hypertension (%)	1394 (58.3)	1416 (88.2)	712 (95.7)
AF (%)	736 (30.8)	744 (46.3)	405 (54.4)
Angina (%)	255 (10.7)	491 (30.6)	366 (49.2)
MI (%)	677 (28.3)	895 (55.7)	519 (69.8)
CABG/PCI (%)	610 (25.5)	893 (55.6)	536 (72.0)
HF hospitalisation (%)	1100 (46.0)	788 (49.1)	361 (48.5)
HF > 5 years (%)	836 (34.9)	643 (40.0)	375 (50.4)
NYHA class III/IV (%)	631 (26.4)	569 (35.4)	341 (45.8)
Ischaemic cause of HF (%)	959 (40.1)	1120 (69.7)	593 (79.7)
Stroke (%)	113 (4.7)	194 (12.1)	159 (21.4)
PVD (%)	61 (2.6)	105 (6.5)	158 (21.2)
Syncope (%)	72 (3.0)	82 (5.1)	77 (10.3)
Gout (%)	131 (5.5)	206 (12.8)	151 (20.3)

SGLT-2 = Sodium glucose transporter, N = Number, SD = Standard deviation, FI = Frailty index, IQR = Inter quartile range, BMI = Body mass index, eGFR = Estimated glomerular filtration rate, DM = Diabetes mellitus, OSA = Obstructive sleep apnea, AF = Atrial fibrillation, MI = Myocardial infarction, CABG = Coronary artery by-pass grafting, PCI = Per-cutaneous coronary intervention, HF = Heart failure, NYHA = New York Heart Association, PVD = Peripheral vascular disease.

## Data Availability

Not applicable.

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
