# Peer review of "The Use of SGLT-2 Inhibitors and GLP-1RA in Frail Older People with Diabetes: A Personalised Approach Is Required"

_metabolites, 2025, doi:10.3390/metabo15010049_

Round 1
Reviewer 1 Report (Previous Reviewer 1)
Comments and Suggestions for Authors
information about new therapies are missing therefore, it useless to say about the quality of article as reader will not get anything new from it. there is a need to define new therapies, selection criteria of these therapies. thus, in current form its not acceptable for publication
Comments on the Quality of English Languagelanguage needs improvement.
Author Response
Thank you for your comments.
Comments: Definition of new therapies.
We have clarified this in the title and across the manuscript. New therapies refer only to SGLT-2 inhibitors and GLP-1RA, no other therapies included.
Reviewer 2 Report (Previous Reviewer 3)
Comments and Suggestions for Authors
The authors have addressed my concerns. No further suggestions.
Author Response
Comment: No changes required.
Response: Thank you.
Reviewer 3 Report (New Reviewer)
Comments and Suggestions for Authors
Dear Authors,
Your article entitled "Defining the characteristics of frail older people with diabetes who should benefit from the new glucose lowering therapies." has been reviewed.
The article deserves attention since it highlights an important topic, related to frailty, which significantly impacts diabetes management in older adults, necessitating tailored approaches based on distinct frailty phenotypes. It is well-justified, addressing the lack of direct evidence for newer therapies like SGLT-2 inhibitors and GLP-1 receptor agonists in this population, while providing practical guidance for optimizing therapy and minimizing risks.
Kindly find below the list of my comments:
1. A little more information could be provided to clarify the U-shaped relationship, the relationship between frailty and body mass index.....
2.Individualized nutritional and hydration strategies in the AM phenotype could be emphasized to avoid further decline in frailty, emphasizing a holistic approach to care.
3.Highlight specific recommendations and strategies that may improve the treatment of these patients, such as combining new therapies with physical exercise and specialized diets to preserve muscle mass.
4.The conclusion could be strengthened by emphasizing the need for future studies to address the gaps in evidence for AM frailty phenotype and to develop tailored therapeutic approaches that prioritize both metabolic and functional outcomes in this population.
The conclusion should be clearer, emphasizing future steps and research directions more specifically, rather than general statements.
Best regards!

Author Response
Many thanks for your comments and suggestions to improve the manuscript.
Comments:
1. A little more information could be provided to clarify the U-shaped relationship, the relationship between frailty and body mass index.
Response: Added, page 6.
2. Individualized nutritional and hydration strategies in the AM phenotype could be emphasized to avoid further decline in frailty, emphasizing a holistic approach to care.
Response: Added, page 14.
3. .Highlight specific recommendations and strategies that may improve the treatment of these patients, such as combining new therapies with physical exercise and specialized diets to preserve muscle mass.
Response: Highlighted, page 12.
4. The conclusion could be strengthened by emphasizing the need for future studies to address the gaps in evidence for AM frailty phenotype and to develop tailored therapeutic approaches that prioritize both metabolic and functional outcomes in this population.
Response: Added, we felt that it suits more in the future perspective section, page 16.
Reviewer 4 Report (New Reviewer)
Comments and Suggestions for Authors
It has been established that in healthy people, with age, there is a decrease in tissue sensitivity to insulin. Concomitant cardiovascular, pulmonary diseases, pathology of the musculoskeletal system lead to inactivity of these individuals. A significant contribution of sarcopenia to the development of insulin resistance was noted. A decrease in muscle mass leads to a deterioration in glycemic control due to a decrease in glucose uptake by muscles, which leads to increased insulin secretion and insulin resistance. While SGLT2 does not offer protection against macrovascular death or renal complications, it is consistently linked to a reduction in overall mortality and the overall burden of cardiovascular complications, including heart failure and cardiac death, in elderly or disabled patients with diabetes mellitus and heart failure. Therefore, the review focuses on the real issue of the glycemic status of older patients with diabetes mellitus and the medication correction of this condition.
The review covers the pathophysiology, kinds of diabetes mellitus presentations in older adults, and the development of therapy strategies. It is written logically. It should be mentioned that almost 90 separate reviews address the issue of diabetes mellitus in the elderly in one way or another, based on PubMed data for the years 2019–224.
Sinclair AJ, Pennells D, Abdelhafiz AH. Hypoglycaemic therapy in frail older people with type 2 diabetes mellitus-a choice determined by metabolic phenotype. Aging Clin Exp Res. 2022 Sep;34(9):1949-1967. doi: 10.1007/s40520-022-02142-8. Epub 2022 Jun 20. PMID: 35723859; PMCID: PMC9208348. - draws attention to the issue of treatment strategies for elderly individuals with diabetes mellitus and its primary phenotypic symptoms, which include anorexia malnutrition and sarcopenic obesity.
Abdelhafiz AH, Emmerton D, Sinclair AJ. New hypoglycaemic therapy in frail older people with diabetes mellitus-phenotypic status likely to be more important than functional status. Diabetes Res Clin Pract. 2020 Nov;169:108438. doi: 10.1016/j.diabres.2020.108438. Epub 2020 Sep 10. PMID: 32920102. - takes into account the various forms of the phenotype of diabetes mellitus in the elderly and evaluates the applicability of treating weaker persons with the disease with hypoglycemic medications.
Abdelhafiz AH, Keegan GL, Sinclair AJ. Metabolic Characteristics of Frail Older People with Diabetes Mellitus-A Systematic Search for Phenotypes. Metabolites. 2023 May 29;13(6):705. doi: 10.3390/metabo13060705. PMID: 37367862; PMCID: PMC10304736. - examines the phenotypic signs and symptoms of diabetes mellitus in older adults.
The authors' integration of the pathogenetic features of diabetes mellitus in the elderly and therapeutic approaches for these individuals makes this review pertinent.
Although half of the reviewed literature had been published more than five years ago, the works are still essentially relevant. The authors state that more research on this issue is necessary and support their conclusion with the material mentioned above. The visual material is easy to understand and provides you with the essential details on this subject in a clear and straightforward manner.
Among the review's drawbacks are the following: - no details regarding the procedures used to choose articles, when publications were included for analysis, important search terms, information databases, etc. - no end-to-end numbering (in numbers) of links;
Line 100: The word "phenotype" needs to be spelled correctly because a letter is missing; the link to the authors needs to be ended with a period (lines 119, 173, 265, 315).
Author Response
Many thanks for your comments and suggestions to improve the manuscript.
Comments:
1. No details regarding the procedures used to choose articles, when publications were included for analysis, important search terms, information databases, etc. - no end-to-end numbering (in numbers) of links.
Response: We have added a new methods section for clarification.
2. Line 100: The word "phenotype" needs to be spelled correctly because a letter is missing; the link to the authors needs to be ended with a period (lines 119, 173, 265, 315).
Response: Corrected, thanks.
Round 2
Reviewer 1 Report (Previous Reviewer 1)
Comments and Suggestions for Authors
its third edition of the article and every time there is a new article and suggestion has not been incorporated. moreover, these suggestions have not been answered / justified. thus quality is as poor as in the first draft of the article.
Comments on the Quality of English Languageneeds revisions
This manuscript is a resubmission of an earlier submission. The following is a list of the peer review reports and author responses from that submission.
Round 1
Reviewer 1 Report
Comments and Suggestions for Authors
The title of the article does not match the content of the article. Authors has listed only few therapies. however, did not provide enough evidence for improvement. The articles are more about the patient characteristics which is helpful for practitioners. But its implementation/ interpretation is vague. Therefore, it is recommended to add more data about novel therapies along with evidence to cure frailty.
Besides, The manuscript is prepared in a classless manner i.e. some lies are incomplete (26) and some are too lengthy to understand.
The similarity index (plagiarism) is too high. thus in current form it is not acceptable for publication.
Comments on the Quality of English Languagelanguage of the article is poor.
Author Response
General Comment
Thank you to all reviewers for their insightful comments. We have tried to accommodate their requests as much as possible. This is a relatively new and complex area, which we have tried to unravel the literature and examine the available evidence. We believe that, following this review process, the manuscript has improved – thank you.
Reviewer 1:
Many thanks for your comments and suggestions to improve this manuscript.
- Title: We have changed the title to closely reflect the content of text, thanks.
- Few therapies of frailty, etc:
A. Just to clarify that this review is a comment to the recent literature, which recommends the indiscriminate use of the new therapies (SGLT-2 inhibitors and GLP-1RA) in frail older people with diabetes showing that they are the most to gain benefit (in terms of CV/renal outcomes, not frailty outcome) compared to non-frail patients.
B. Our manuscript, therefore, reviews such claimed evidence and explored that it applies only to frail older people who are at least have normal weight. We demonstrated that the benefit was most in frail people who are obese, or a sarcopenic obese but not in all frail older people as the literature suggests.
C. This review confirms our previous publications in this topic, which suggest that frailty is not one homogenous population but spans across a spectrum with different metabolic characteristics. The sarcopenic obese (SO) frail phenotype at one end, which is likely to benefit from the new therapy and the anorexic malnourished (AM) phenotype at the other end, which is not likely to benefit due to high risk of side effects of further weight loss, dehydration, hypotension and falls.
- Implementation is vague: We regret that you feel it is vague. We have reviewed the take home messages and made it clearer in the conclusion section and the key points.
- Review is classless: We understand that it may look like this, but it is in the first place a review and a comment on the little available current literature, which suggests the indiscriminate use of the new therapy in frailty. Frailty was mistakenly seen as one homogenous category. Therefore, this literature recommendation is inaccurate and our manuscript was just to highlight this point.
- Lengthy lines: We have reviewed this and tried to shorten sentences.
- Plagiarism: There may be some overlap with our previous publications in the same topic. We have reviewed this as possible, but will be happy to review again if index is still high.
We have added new references across the manuscript to further support our findings.
Please note that additions are highlighted and deletions are struck through.
Reviewer 2 Report
Comments and Suggestions for Authors
The review study submitted for peer review aims to find new directions in understanding metabolic phenotypes of frailty and aimed primarily at helping specialists in the field of medicine in their daily decision-making for the use of the new therapies in the high-risk group of patients with frailty.
The study aims to investigate the problem of complicating factors of diabetes in the elderly, studying various aspects of traditional and new methods of treatment in this case. The results of the study show that frailty is an important emerging complication of diabetes in older people and there is no direct evidence to suggest the benefits of the new therapies of SGLT-2 inhibitors and GLP-1RA in this group of patients. The current available evidence is indirect and showed that the SO frail phenotype of older people with diabetes to benefit from this therapy. The unfavourable metabolic profile of this phenotype and the highly prevalent cardiovascular risk factors put this group of patients at a high baseline cardiovascular risk and therefore are the most to benefit from such therapy. Therefore, according to the authors, based on the study, clinicians should consider the early use of the new therapies in frail older people with diabetes who are normal or overweight with prevalent cardiovascular risk factors.
Thus, the authors make practical recommendations that are significant for medical science: the new therapies of SGLT-2 inhibitors and GLP-1RA are effective in frail older people with diabetes; sarcopenic obese frailty phenotype is the likely frailty phenotype to benefit from such therapy, because of the unfavourable metabolic profile and high baseline cardiovascular risk of the sarcopenic obese phenotype, it stand to benefit most of such therapy.
Despite the logicality of the article, theoretical and practical significance, it is necessary in the discussion and conclusions of the study to clearly indicate the uniqueness of the results obtained in comparison with the studies of other scientists. It is necessary to expand and systematically adjust the section with key research points in the final part of the article.

Author Response
Reviewer 2:
General Comment
Thank you to all reviewers for their insightful comments. We have tried to accommodate their requests as much as possible. This is a relatively new and complex area, which we have tried to unravel the literature and examine the available evidence. We believe that, following this review process, the manuscript has improved – thank you.
Many thanks for your comments and suggestions to improve this manuscript.
To clarify the message of this review: We have summarised findings and clarified the clinical message of this manuscript in the conclusion section and the take home key points.
Reviewer 3 Report
Comments and Suggestions for Authors
The article explores the issue of frailty caused by complications of diabetes related to cardiac and renal function; however, the introduction of this section is insufficient. It merely states that the new therapy improves frailty resulting from cardiac and renal complications of diabetes, inferring its impact on diabetes treatment without directly demonstrating it. I recommend revising for publication, with specific comments as follows:
- In the Introduction, the authors emphasize that the new therapy can improve cardiac and renal function, leading to the inference that this therapy could be used to treat diabetic patients, but there is no direct connection established. Please clarify this.
- I suggest adjusting the order of Chapters 2 to 5; the logic of the article should analyze the reasons first, followed by treatment. Recommend changing to: Frailty and diabetes—Patients’ characteristics—Frailty metabolic spectrum—New diabetes therapies in frailty.
- In the Frailty and diabetes section, the authors discuss frailty caused by cardiac and renal dysfunction, but this part does not illustrate the relationship between frailty induced by cardiac and renal dysfunction and diabetes.
- In the Frailty metabolic spectrum section, I recommend creating a U-shaped frailty metabolic spectrum diagram to enhance reader experience.
- Continuous glucose monitoring is crucial for diabetes treatment; please elaborate on advancements in this area, such as in Biosensors and Bioelectronics, 2024, 244: 115822.
- The word "therefore" at the end of the abstract is unnecessary; please remove it.
Author Response
General Comment
Thank you to all reviewers for their insightful comments. We have tried to accommodate their requests as much as possible. This is a relatively new and complex area, which we have tried to unravel the literature and examine the available evidence. We believe that, following this review process, the manuscript has improved – thank you.
Reviewer 3:
Many thanks for your comments and suggestions to improve this manuscript.
- Cardio-renal benefits of new therapies: We have clarified this in the introduction section.
- Adjusting sections: We have moved sections as suggested. We left patients’ characteristics section last as it describes patients included in the new therapies studies, which should come first.
- Relation between frailty and cardio-renal complications: expanded as suggested in the section of frailty and diabetes.
- Frailty diagram: We have added Figure 1, thanks.
- Continuous glucose monitoring: We have added this reference as suggested.
- Word therefore in abstract: Removed.
New references added across the manuscript to provide more support of our findings.